# ARMAX Forecast Model for Estimating the Annual radon Activity Concentration in Confined Environment by Short Measurements Performed by Active Detectors

**DOI:** 10.3390/ijerph19095229

**Published:** 2022-04-25

**Authors:** Andrea Pepperosa, Romolo Remetti, Francesca Perondi

**Affiliations:** 1CAEN SpA/Spectroscopy Division, Nuclear Waste Management (NWM), Decommissioning and Dismalting (D&D), Via della Vetraia 11, 55049 Viareggio Lucca, Italy; a.pepperosa@caen.it; 2Department Basic and Applied Sciences for Engineering, Energetic and Nuclear Engineering, Sapienza University of Rome, Via Antonio Scarpa 14, 00161 Rome, Italy; romolo.remetti@uniroma1.it

**Keywords:** indoor radon activity concentration, radon detection, forecast models, continuous radon monitor, MATLAB, ARMAX series, historical series

## Abstract

This work aims to implement a forecast model that, combined with the use of active instrumentation for a rather limited time, and with the knowledge of a set of data referring to the environmental parameters of the place to be monitored, can estimate the concentration of indoor radon activity for longer time periods. This model has been built through the MATLAB program, exploiting the theories of time series and, in particular, ARMAX models, to reproduce the variation in the concentration of radon activity. The model validation has been carried out by comparing real vs. simulated values. In addition, analytic treatment of input data, such as temperature, pressure, and relative humidity, can reduce the influence of sudden transients allowing for better stability of the model. The final goal is to estimate the annual radon activity concentration on the basis of spot measurements carried out by active instrumentation, such to avoid the need to measure for an entire calendar year by the use of passive detectors. The first experimental results obtained in conjunction with active radon measurement demonstrates the applicability of the method not only for forecasting future average concentrations, but also for optimizing remedial actions.

## 1. Introduction

Recently, European national laws on radiation protection have focused their attention on *existing exposure situations* [1], in particular with regard to radon gas. Exposure due to the inhalation of radon gas and its progeny has been recognized as the second cause of lung cancer from the World Health Organization (WHO) [2]. This is a consequence of the estimation of *Lifetime excess of absolute risk* (LEAR) of lung cancer associated with concentrations of radon and its progeny updated by ICRP 115 [3]. This value has been doubled on the basis of new cohort studies on miners in ICRP 103 [1], and on the basis of new predictive models proposed by BEIR VI [4], that can project the relative risk obtained by a certain follow-up time, to an entire lifetime period. Thanks to these models, to the larger time extension of the Life Span Study (LSS) [5], and to the revision of *detriment* values [1], the internal exposure conversion factors have been updated on this basis. In particular, ICRP 137 [6] proposes the conversion factors (CF) of 20 mSv/WLM for workers and 10 mSv/WLM for miners by using epidemiological studies and new biokinetic and dosimetric models. The new concept of risk associated with the inhalation of radon and its progeny has caused the introduction of a so-called Reference Level (RL) that has substituted the old Action Level (AL) [7]. The RL is a value above which it is necessary to implement remedial actions and under which it is necessary to lower the radon activity concentration As Low As Reasonably Achievable (i.e., the ALARA concept derived from the Optimization Principle of Radiation Protection) [1].

The European Union’s international landscape regarding exposure to natural radioactivity is placing increasing attention on the radon problem [8]. Such increased sensitivity, derived from the previously described current view on the matter, caused substantial modifications of the national laws of individual EU member states. Thanks to the improved knowledge of the health risks related to indoor exposure to radon gas, and in particular to its progeny [9], the European reference levels have suffered a sharp drop down in the last 10 years. Further, the frequency of measuring indoor radon activity concentration has also increased significantly by international directive and national laws [10,11].

New issues have to be faced up to now, for example, when selling or purchasing a dwelling. In such a situation, it will be necessary to implement the documentation of the property with a radon technical report. Obviously, in such situations, there is no time to wait one year to determine the annual radon concentration.

Many measuring devices are currently available, and these instruments are suitable for different purposes. Passive instrumentation is the only one currently able to provide averaged results throughout the year.

This causes a certain difficulty in carrying out extended surveys on medium-high geographical scales and in the timeliness with which it is possible to identify critical areas and implement prevention actions, or remedial actions, where necessary [12]. As a consequence, there is a need to develop forecasting models for estimating the concentration of this gas in residential and working environments [10,13].

## 2. Materials and Methods

Passive instrumentation, commonly based of Solid-State Nuclear Track Detectors [13], is currently the only one allowing the determination of the average annual activity concentration [14,15]. These devices provide very reliable results, but at the expense of the measurement time needed, which covers precisely the entire calendar year [15,16,17].

It would be interesting to think of combining active instrumentation, which provides real-time data, with predictive models to correlate the variability of the concentration to environmental parameters, such as temperature, pressure, and relative humidity (derived from historical data on the analyzed area).

In this way, it would be possible to estimate the annual average concentration by using models such as the one proposed hereafter, and by committing active instrumentation for a period of time limited to a few weeks.

The present work is based on the assumption that the indoor variability of radon activity concentration can be thought of as a sequence of random variables (i.e., a random vector of infinite size).

In other words, the method is based on the collection of a sample of data which is sufficiently representative of the dynamics envisaged, and on the identification of the best theory to extrapolate a forecast model. Such a model will have enough “memory” of itself, and will therefore reproduce the real phenomenon, to predict its evolution over time.

### 2.1. Mathematical Model

The mathematical model belongs to the Black Box family, a mathematical tool unsensitive to the physics of the process to be analyzed [18,19].

Even if the physics of the production of radon from ^226^Ra is known, as well as its penetration into the indoor environment, there is no universal physical law that can be used to assess the correct and punctual concentration of radon activity inside a room: it can only be measured [15].

The mathematical model is a bridge between signals and systems.

The Black Box model receives u_i_ input variables and gives as output $y_{i}=f\left(u_{i}\right)$ [19].

If the model is dynamic, the output will depend on the past history of inputs and outputs. In other words, the knowledge of the past allows to obtain an estimate of the current situation.
(1)$$y_{\mathrm{now}}=f\left(y_{\mathrm{old},\;}u_{\mathrm{old}}\right)$$

By using the ARMAX model theory [20,21,22], the time evolution of variable y can be expressed as follows:(2)$$y\left(t\right)=\frac{B\left(q\right)}{A\left(q\right)}u\left(t\right)+\frac{C\left(q\right)}{A\left(q\right)}e\left(t\right)$$
where:y(t) is the variable we want to know: future indoor radon concentration;u(t) is the input variable (input data) and is measurable;e(t) is the error (or disorder) and it is not observable. It represents the difference between the reality and the simulation proposed by the system. It can only be estimated retrospectively, reversing Equation (2);The fraction B(q)/A(q) is a rational function (relationship of polynomials) in the variable q (that is the adaptive regression parameter) and represents the so-called transfer function of the deterministic part of the system, between exogenous input and output;The fraction C(q)/A(q) represents the transfer function of the disruptive part of the system, between exogenous input and output.

The Equation (2) allows to see the variable y(t) as a transformation of the variables u(t) and e(t).

Its structure can be seen in the flow chart, presented in Figure 1.

This model is linear in input, output, and error variables, but not linear in regression parameters. The outputs are calculated as the scalar product of the regression vector for that of the parameters:(3)y(t)= q × φ(t, q)

The error term is able to take into account the non-linearity of the process, the non-measured noise, and noise in the measurements.

The determination of parameters q requires a procedure of non-linear minimization (for example, at least squares), which is the one that determines the most time of calculation and also the presence of the error, and requires the measurement in the field of the output variables (of which e represents precisely the deviation of the model).

Implementation of the model can be summarized in the following steps:The number of input and output variables and the range of variability of the same are determined;The sampling time is defined;The length of the regression vector, the order of the model vs. each variable, the linearity, or non-linearity of the model vs. the regressors are defined;The algorithm for the determination of the parameters is chosen by a deterministic procedure (minimization of the error committed);The predictive ability is tested through a set of unused data (cross-validation set) carefully selected and distinguished from the training set (learning set).

In general, the sampling time $t_{s}$ is 5–20% of the typical system time. If this is too long, the dynamics of the process may not be identified; on the contrary, if it is too short, there is a risk of increasing noise sampling, excessive manipulation of data, and interference with the identification algorithm [23].

The input data, before being used, can be pre-treated, for example, through mathematical operators that break down the excessive oscillations (moving average), or through filters high-cut or low-cut, to delete sudden oscillations [23].

The parameters q of the model are determined by a regression procedure, which in the case of deterministic signals, is equivalent to solving:(4)$$\underset{q}{\min}\left\{\sum_{i=1}^{n_{y}}\sum_{t=1}^{n_{s}}[y_{i}^{\mathrm{real}}\left(t\right)-f_{i}\left(\varphi \left(t\right),q\right){]}^{2}\right\}$$
where $n_{y}$ is the number of output variables and ${\;n}_{s}$ is the number of samples.

### 2.2. Model Applied to Radon Case

The ARMAX model, notwithstanding being developed for financial applications, can be applied to the radon case because it is characterized by the following features:Trend: Monotonous, long-term tendential movement that highlights a structural evolution of the phenomenon. This is due to causes that act in a systematic way; for instance, the degree of fracturing of the subsoil, its geological origin, the missed sealing of cracks, the emanation from certain building materials, and so on [8].Cycle: Movement, or cyclical fluctuation, originating from more or less favorable conditions of expansion and contraction of the phenomenon; for example, the daily cycle caused mainly by the opening of the windows in the morning, or the activation of the heating or ventilation systems [9].Seasonality: Fluctuations caused by climatic factors. As it is well known, the higher concentration of activity is expected in winter, rather than in summer. However, it is always necessary to keep in mind the particular cases of inverse seasonality, deliberately excluded from this study [24,25].

The input data of the system are the variables that influence the indoor concentration of radon gas: pressure, temperature, humidity, and wind speed [26].

Figure 2 shows the design of the model implemented.

On the left of Figure 3, it is possible to see the variables of interest, (C_Rn_, P, T, U, W) and the relative structural parameters of regression, i.e., regressors (n_a_, n_b_, n_c_). On the right, there is the ARMAX process used for the creation and also for the validation of the model itself in two different time windows, (t_0_ < t < h) and (t > h). If you take, for example, a measuring period of 500 h, during the first 200, the degrees of the parameters are estimated, and during the subsequent, the model is verified. It means that two different sets of data, about C_Rn_ measured and environmental parameters, are used to create and validate the model. The best order of the regressors is chosen by the NET criterion (NET box in Figure 3) that impose the mean value of C_Rn_ simulated to be as close as possible to the mean value of C_Rn_ measured. The area between the curve of the simulated concentration and that of the measured concentration has to be the minimum possible (criterion of least squares, as mentioned above). This is equivalent to calculating the second moment of the difference between the values obtained from the simulation and the values actually measured (4). For clarity, the environmental parameters are represented by P = pressure, T = temperature, U = humidity, W = wind. At the start of the process, a maximum degree is set for each variable and the program will perform for as many cycles as there are possible combinations of degrees and will record the results obtained in a matrix in which each dimension represents a degree of freedom of the model. Through the realization of a special function, the value that satisfies the chosen criterion is extracted, and the program shows the exact position of the element in the matrix. This position corresponds to 6 indices which represent the values to be assigned to each independent variable of the model. At the end of the process, in addition to the simulation graph, which provides a purely statistical indication of the evolution of the phenomenon, the program shows another type of graph, relative to the goodness of the model (FITTING) providing information on the accuracy of the model itself according to the number of experimental points. It is important to compare all the results to verify its robustness.

## 3. Experimental Results

The active device used to measure the radon activity concentration is an AlphaGUARD PQ-2000 pro [28], both for collecting the learning set and the validation set. It was used for a limited period of time (five weeks) to obtain the set of data to be inserted into the model. Obviously, every active detector can be used for this purpose, but it is important that the device can store all the records, for all of the time needed. The monitored environment was the Radiation Protection Laboratory, situated in the Department of Basic and Applied Sciences, in “Sapienza”—University of Rome. The data about the environmental parameters were collected from the Integrated Agrometeorological Service of Lazio [29] and measured in a station situated near the monitored room. The data were collected from March to April of 2021.

In Figure 4, Figure 5 and Figure 6 some results are presented.

In Figure 3a, we have a simulation blue line, and a green line of real data. The simulation has been created by a measuring time of 400 h. In Figure 3b, we have the simulation created with a measuring time of 600 h. It can be noted that after an initial transient, the model tends to align with reality, and this is more evident with the increase in monitoring time. Figure 3c,d show the same results as the 3a and 3b figures but in the details (zoomed). This is the first simulation carried out and just a fraction of the collected data was used (just sixteen days of monitoring). For this reason, a poor convergence of the simulation with the reality was expected. Notwithstanding, the simulated results show a satisfactory convergence with the real ones after about 400 h. For this reason, it was decided to add a wider data set.

The sampling time was increased, assuming 800 and 1000 h of monitoring, as shown in Figure 4. Figure 4a,c were obtained using data collected in 800 h of monitoring, while Figure 4b,d concerned 1000 h. The same trend, previously identified in Figure 3, was also obtained in this case, but with a better agreement.

After a natural period of oscillation, which is a characteristic of the dynamic systems [20], the simulation follows the real course quite well, notwithstanding a certain bias. In this case, the bias is smaller than in the previous simulation because of the increase of available data. In the Figure 3 simulations, relative bias, quantified by the percentage deviation between the average values, was about +67% at 400 h, and +56% at 600 h. Instead, in the Figure 4 simulations, relative bias was about −25% at 800 h, and −12% at 1000 h.

Figure 5 shows how the simulated data fit the experimental ones.

This figure shows the fittings of the model for all the proposed simulations (with a different number of creation points). The black line is the measured variation of activity radon concentration and the pink line is the simulation.

Figure 5a, relative to h = 400, is the only one that does not exhibit good results, probably due to some noise on the input data. In all of the other cases, the fitting is, in general, very good. Making the hypothesis that noise could be caused by abrupt variations of environmental data, it was decided to implement noise correction, as discussed below.

The Figure 6 shows the trend of the measured environmental parameters (pressure, relative humidity, temperature, and wind speed) collected by the meteorological station, and refers to the same period of radon data collection.

It was decided to operate a “flattening” of the input data by eliminating the time intervals showing abrupt variations that are particularly evident where relative humidity and temperature are concerned. As shown hereafter, in this way, noise was reduced but, unavoidably, the experimental data on radon concentration were reduced, so it was necessary to increase the monitoring time.

Figure 7 and Figure 8 show the simulation and measurement trends before and after the environmental data treatment.

The improvement of the simulation is evident by observing the graphs on the right in the previous figures. The bias is reduced and the model estimates in a better way the variation of indoor radon concentration. In the best simulation (Figure 8b), the bias is about +11%.

For testing the chance that the environmental data flattening could worsen the agreement of the simulation with reality, new fittings have been performed, as reported in Figure 9.

The goal of this test was to verify the influence of some noise from the environmental data on the simulation, and the goodness of the model. It is clear that the operation of flattening of input data improves the simulation and, at the same time, it does not influence the goodness of the model: the fitting is the same before and after the treatment, and it obviously improves with the number of model creation points (the hours of monitoring). These results have demonstrated that it is possible to forecast the variation of indoor radon concentration because this phenomenon has a certain memory of itself, i.e., its own history, and follows such evolution over time, but it is necessary to exclude some abrupt variations in the parameters that affect the phenomenon itself.

## 4. Discussion

The results are rather encouraging, and inaccuracies are mainly due to the quantity and quality of the data. In order to estimate the parameters contained in the model, it is necessary to have available series of measurements of all the input and output variables for a sufficiently long time period. However, the initial target of estimating radon concentration by spot measurements with active instrumentation seems to be achieved. For ameliorating performances of the model, some improvements could be tested. In particular, it could be possible to consider ambient parameters, in addition to environmental ones, which are susceptible to influence the phenomenon, such as the degree of burial of the room, the air exchange rate, and the contribution of building materials. An important improvement could be achieved by also considering the pressure difference between the indoor and outdoor environments. Anyway, the basis has been laid for estimating the average annual concentration of radon gas through active instrumentation by the development of a forecasting model. In such a way, it will be possible to reduce the monitoring time for assessing radon concentration: this is the main goal of the study, and it would be very helpful in many practical situations. This could be a method of interest when it is necessary to quickly estimate the average annual concentration, for example, when radon certifications are needed when selling or purchasing dwellings. Typical SSNTD must remain in place at least some months, and then they have to be sent to the laboratory for analyses. This implies longer times to obtain an average value of radon concentration in a dwelling. Instead, by using spot time measurements and the forecasting model implemented on a computer, the technician asked to assess the radon situation could produce technical reports in acceptable times. The model could also be very helpful when designing remedial actions. In effect, the results of passive detectors do not allow to appreciate the time evolution of radon concentrations. Instead, this is fundamental when installing devices such as depressurization well systems. When designing such remedial actions, it is useful to measure the time evolution of the indoor radon concentration by active detectors for some days. This is important to estimate the deepness of the underground layers that mainly contribute to radon emission, in relation to the speed of increasement of the indoor concentration. Further, it is fundamental to appreciate the variation of the concentration vs. some factors, such as heating or ventilation systems, habits of tenants, etc. By using active detectors for the same time (some days) and by implementing these measures with the model proposed in this study, it would be possible to have more available data, to appreciate in the best way the radon situation, and to design the best remedial actions. Further, when testing the effectiveness after having carried out these remedial actions, it is currently necessary to follow the radon concentration for some months. Instead, by using the model in conjunction with active detectors used for some weeks, the monitoring time could be significantly reduced.

## 5. Conclusions

By using the historical series theory, and in particular the ARMAX model implemented in MATLAB, the model has been able to foresee future concentrations. Such a capability has been verified by fitting real data obtained by active instrumentation. The input data were radon measures collected by an AlphaGUARD and environmental parameters given by meteorological stations regarding the place to be monitored, such as pressure, temperature, relative humidity, and wind speed. Further, better results on forecast capabilities were reached by eliminating transitories of environmental data.

This work demonstrated that it is possible to build a forecasting model that, used with active measurement devices, can provide an estimation of the variation of indoor radon activity concentration. Even if other forecasting models are available in the literature [30] the novelty of this study is to be used in conjunction with active instrumentation. Once implemented on a computer, this model can be used “in-field” to give immediate results that allows to estimate average radon concentrations in dwellings, to design remedial actions, and to evaluate their effectiveness. In this way, observation times could be reduced, intervention procedures could be simplified, and, in addition, Reference Levels could be more effectively prevented from being exceeded [17].

## Figures and Tables

**Figure 1 ijerph-19-05229-f001:**
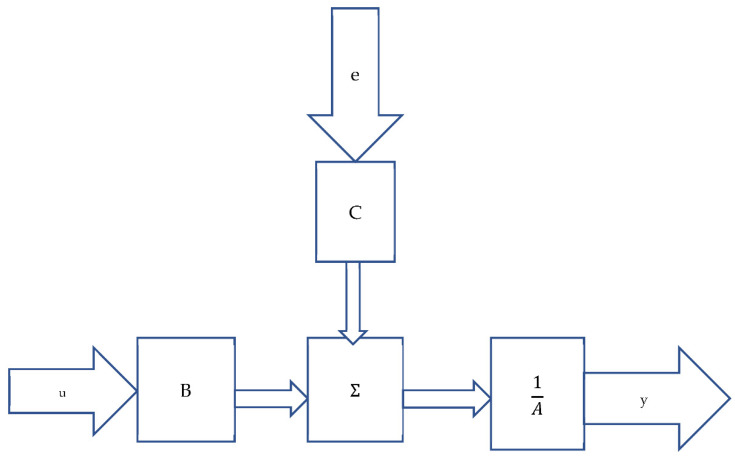
Flow chart of the ARMAX models.

**Figure 2 ijerph-19-05229-f002:**
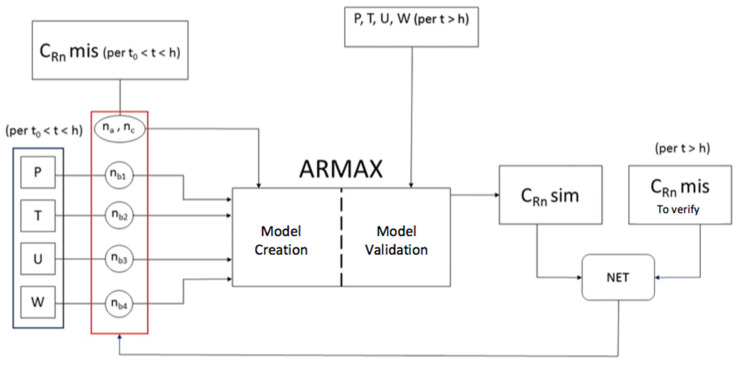
Structure of the ARMAX model to be used with MATLAB software (The Mathworks, Inc., Natick, Massachusetts, USA) [27].

**Figure 3 ijerph-19-05229-f003:**
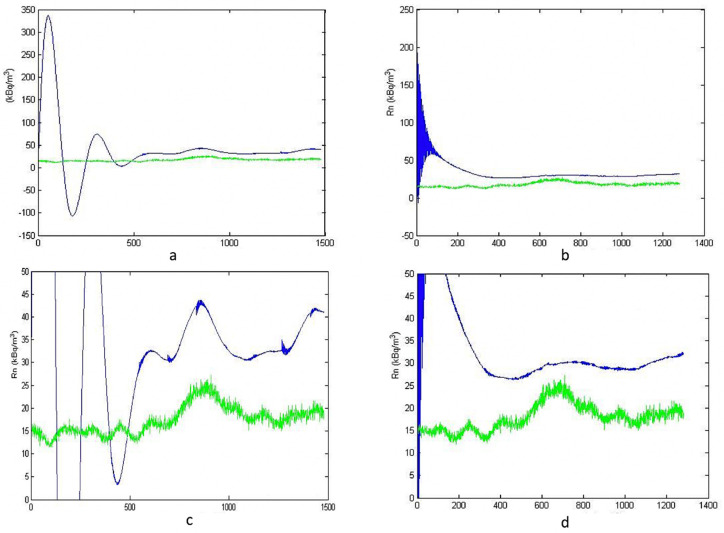
Simulation vs. reality for 400 (**a**) and 600 (**b**) hours. The graphs are zoomed, respectively, in figure (**c**,**d**).

**Figure 4 ijerph-19-05229-f004:**
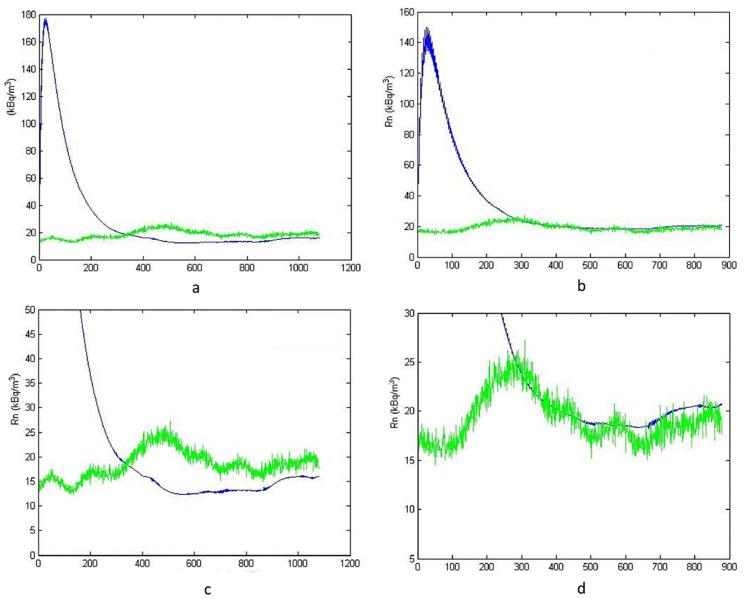
Simulation vs. reality for 800 (**a**,**c**) and 1000 (**b**,**d**) hours. Figure (**c**,**d**) are zoomed views.

**Figure 5 ijerph-19-05229-f005:**
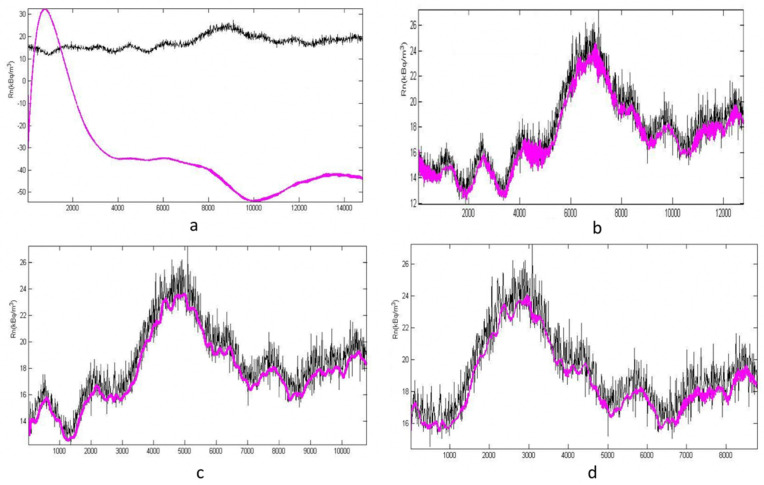
Fitting for the simulation with h = 400 (**a**), 600 (**b**), 800 (**c**), 1000 (**d**).

**Figure 6 ijerph-19-05229-f006:**
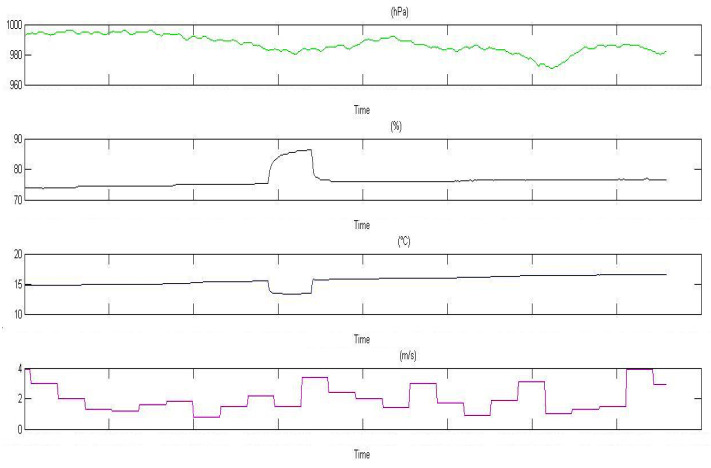
Trend of the measured environmental parameters. From top to bottom: pressure; relative humidity; temperature; wind speed.

**Figure 7 ijerph-19-05229-f007:**
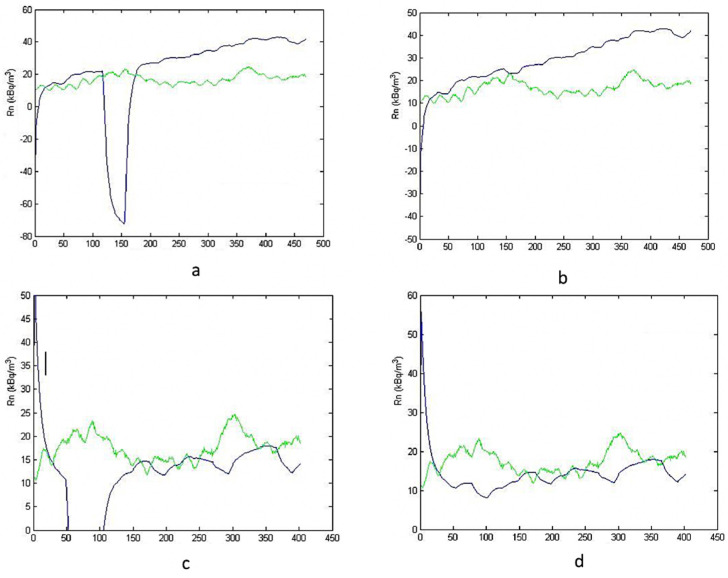
Influence of noise on input data. (**a**,**c**) before, and (**b**,**d**) after data treatment. (**a**,**b**) for 100 h, and (**c**,**d**) for 168 h.

**Figure 8 ijerph-19-05229-f008:**
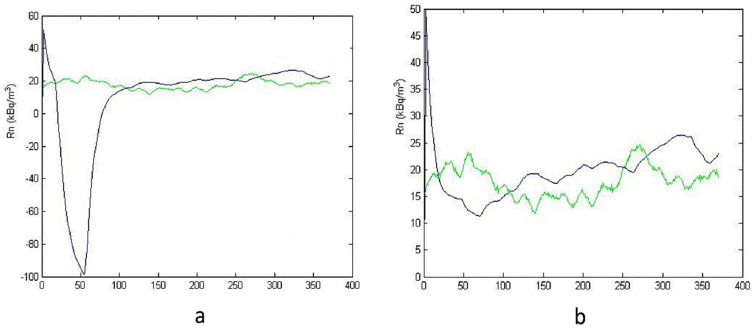
Influence of noise on input data for h = 200. (**a**) before, and (**b**) after data treatment.

**Figure 9 ijerph-19-05229-f009:**
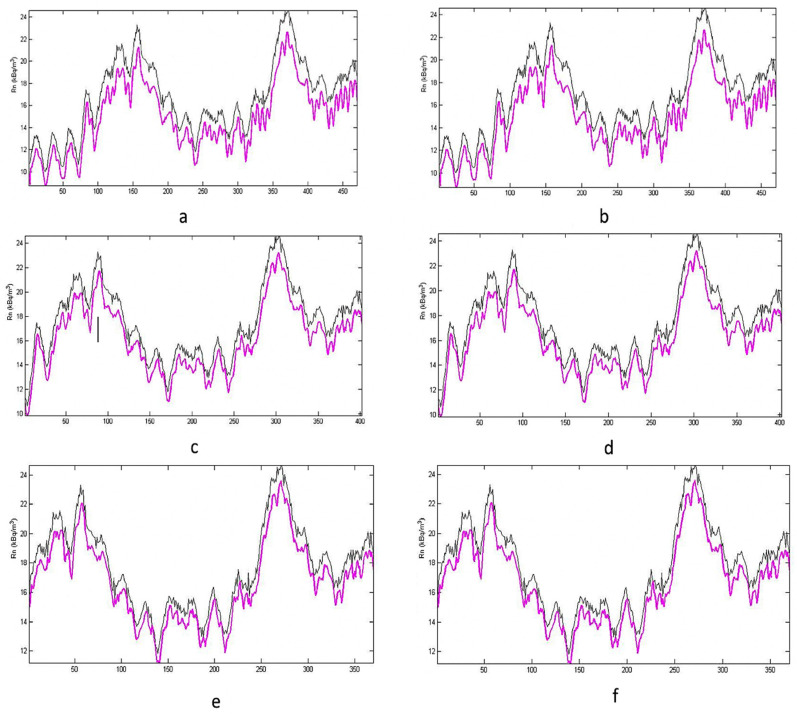
Fitting after flattening of input data for a different number of model creation points: h = 100 ((**a**): before, (**b**): after); h = 160 ((**c**): before, (**d**): after); h = 200 ((**e**): before, (**f**): after).

## Data Availability

https://www.siarl-lazio.it Last accessed on 12 April 2022.

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
