# Peer review of "ARMAX Forecast Model for Estimating the Annual radon Activity Concentration in Confined Environment by Short Measurements Performed by Active Detectors"

_ijerph, 2022, doi:10.3390/ijerph19095229_

Round 1

Reviewer 1 Report

Dear authors,

Some comments and suggestions are listed below. 

Manuscript ID: ijerph-1631696

Title: ARMAX forecast model for estimating the radon annual activity concentration in confined environment by short measurements performed by active detectors

The presented study is of interest in the area since it shows the implementation of a model to estimate radon concentrations over time and the relationship of these Rn concentrations with climatic variables such as pressure, temperature, and humidity. However, the way it is presented is hard to read and therefore challenging to understand.

As an essential review and to improve the manuscript, some suggestions are the following:

  1. Write each section of the article according to the journal's guidelines. Punctuation signs, mainly commas, are worst written, as well as the paragraphs do not reflect the continuity of writing and explanation of each one, for instance, in the introduction and methodology sections.
  2. Modify the description of the mathematical model concisely and clearly, describing the meaning of each variable; it should be clear to readers how each mathematical variable correlates with the measurement variables.
  3. Describe the Spatio-temporal characteristics (location and season), where the radon and climatic measurements were performed, since the results are presented without a description of the area that supports their behavior.
  4. Result section. Do the radon concentration values ​​measured concerning time (hours: 400, 600, 800, etc.) behave statistically according to a normal distribution? If not, as recommended, they are treated statistically with a data transformation (lognormality).
  5. It is recommended to improve the quality of the figures and refer them with numerators (a,b,c, etc) according to the standards established by this journal. Once it is improved, a broad discussion of those results is needed.
  6. Improve the discussion section. Please explain why this model would be helpful to anyone who wants to implement it. The authors should ask themselves and describe the advantages of using this estimation model compared to other existing ones; how does it improve current knowledge? Make a deeper discussion of the results.
  7. Conclusion section. This section should be presented with the most important findings; the way is currently written is very similar to the manuscript's abstract.
  8. The references section will increase in recent citations once the discussion of results is broader and more profound.

Reviewer 2 Report

The article " ARMAX forecast model for estimating the radon annual activity concentration in confined environment by short measurements performed by active detectors" presented by Andrea Pepperosa and co-authors is unfortunately without interest and cannot be useful to the related scientific community. In my opinion, this article should not be accepted in the International Journal of Environmental Research and Public Health.

Authors used very weak scientific English to describe their work. Introduction to their research is not good. They did not use enough literature research, and the used references are inappropriate.  Figures are not well prepared.

Reviewer 3 Report

Referee report on

ARMAX forecast model for estimating the radon annual activity concentration in confined environment by short measurements performed by active detectors

Written by Andrea Pepperosa et al.

This ms is well done and well written, and I recommend it for publication.

Some reduction of text in materials and methods is needed, because there are too many unnecessary (for the present problem and work) information. Some citation on ARMAX method would be probably enough.  

So far, 3 flow charts are presented, I think at least one could be deleted (maybe first one). 
Also, description of Alpha Guard is not needed here, it is very well known and widely used device.

Reviewer 4 Report

Generally, I can conclude that the article is very interesting.

The authors describe a novel system dedicated to the prediction of indoor radon concentration. The system requires an active radon monitor for preliminary measurements and it allows to predict the future radon concentration taking into account various environmental conditions.

The article is well written and composed. The introduction describes all needed facts and data. The description of the system is clear and understandable. Also, conclusions are satisfying.
Nevertheless, I have found some minor issues:
line 117 - missing bracket ")",
line 392 - the description of figure 10 is not clear

Concluding - I would recommend to accept with above corrections.

Author Response

The missing bracket has been added and the description of all figures has been improved. 

Round 2

Reviewer 1 Report

Thanks, authors, for the responses and the new version of the manuscript.

A better explanation of the advantages and importance of this model over others must be provided; similarly, how this model improves current knowledge. 
Make a deeper discussion of the results, not only by describing them.
I did not find the lines 1871 – 1881.

Also, the abstract must be written according to the modifications on results and discussion.

Remove the dots in the middle of the figure numbers (e.g., 4. a).

Reviewer 2 Report

I do thank the authors for the efforts they did to improve the quality of their article. Even though, the following remarks must be token in consideration:

In the scientific and official writing, it is forbidden to use apostrophe/ contractions.

“It’s” is incorrect, and must be replaced by “it is” in the text:

Page 2, Line 43 and line 44 - Page 3, Line 106 - Page 5, Line 172 - Page 5, Line 194 - Page 6, Line 204 -Page 10, Line 349 - Page 11, Line 372 - Page 11, Line 388

In the same context, “doesn’t” is incorrect and must be replaced by “does not”: Page 10, Line 350

Page 2, line 88: Black Boxe family (black Box?)

Page 5, Line 175: “If you take for example …” the sentence should be reformulated. It can be written in a passive voice, the part “if you take for example” can be removed, or “we” can be used instead of “you”.

Page 5, Line 175: (t0< t <h) you must decide if you want to add a space before and after the sign “<” everywhere.

Page 6, Line 209: “Data were collected from March to April” please specify the year.

Page 6, line 203: "few weeks" how many weeks exactly?

Finally, I highly recommend authors to try a language editing service or to get help from someone to improve their paper writing.
